# Beyond Static Bias: Quantifying Fairness Variability in CheXpert

**Ines Ayed**
Department of Mathematics and Computer Science
University of the Balearic Islands
07122 Palma, Spain
ines.ayed@uib.es

**Gabriel Moyà Alcover**
Department of Mathematics and Computer Science
University of the Balearic Islands
07122 Palma, Spain
gabriel.moya@uib.es

**Fernando Alonso-Fernandez**
School of Information Technology
Halmstad University
Halmstad, Sweden
feralo@hh.se

**Antoni Jaume-i-Capó**
Department of Mathematics and Computer Science
University of the Balearic Islands
07122 Palma, Spain
antoni.jaume@uib.es

## Abstract

Fairness in machine learning is typically assessed through static point-estimate metrics that overlook the robustness and reliability of model behavior under biased data. We introduce a statistical framework to analyze the relationship between the variability of dataset bias and the variability of a model's fairness gaps. Using Monte Carlo simulation, we quantify bias in the CheXpert dataset and find that while bias is small, it is consistently stable with near-zero variance across its five most common pathologies. Applying a mixed-effects model, we then examine how this stable bias relates to fairness variability in leaderboard models. We find that model fairness can fluctuate unpredictably even when dataset bias is modest but stable, revealing a hidden robustness failure in fairness evaluations. Our results underscore the need to move beyond static fairness metrics toward evaluation methods that explicitly characterize robustness under subpopulation and distribution shifts, aligning with the broader goals of building reliable machine learning under imperfect data.

## 1 Introduction

Machine learning (ML) algorithms have reached high accuracy in medical image diagnosis, making them a great tool to democratize medical access and improve the accuracy of disease diagnosis [1]. However, research points to the fact that these algorithms are under-diagnosing certain demographic groups, where the AI algorithm may incorrectly identify a person with a disease as healthy, potentially delaying necessary medical care [2, 3]. This compromises their reliability in practice from a fairness perspective.

39th Conference on Neural Information Processing Systems (NeurIPS 2025) Workshop: Reliable ML from Unreliable Data.

For example, a study by Seyyed et al. [2] showed that AI models produce significant differences in the accuracy of automated chest x-ray diagnosis across racial and other demographic groups such as females or Black patients, even when the models only had access to the chest x-ray itself. Similarly, Larrazabal et al. [3] found a consistent decrease in performance of AI systems for computer-assisted diagnosis for underrepresented genders in training datasets. With the increasing deployment of these algorithms in real-world applications, there is a growing fear of mirroring or amplifying human bias in typically marginalized groups.

One potential source of algorithmic bias is the bias present within the training dataset. Thus, it is important to analyze the demographic bias in datasets used in medical imaging tasks. While public datasets from various regions worldwide are becoming more accessible, there remains a critical lack of standardized metrics to assess and quantify demographic bias in these datasets [4]. Without such metrics, it is difficult to determine the extent to which biases exist and how they may impact model performance.

We hypothesize that dataset bias and fairness are related. To investigate this, we introduce a framework combining Monte Carlo simulations and mixed-effects models to systematically quantify the relationship between dataset bias and model fairness metrics. Our simulations reveal that dataset bias is consistent and stable, suggesting it is an intrinsic feature of the CheXpert dataset rather than a sampling artifact. In contrast, fairness gaps can be highly variable across resampled datasets.

Mixed-effects modeling further shows that the impact of dataset bias varies by metric. Demographic Parity Gap and Equalized Odds Gap are more sensitive to overall dataset bias (as measured by Cramer's V; a measure of association between demographics and pathology labels) than Equal Opportunity Gap, indicating that the relationship between bias and fairness is not uniform. Additionally, model variability contributes less to fairness variability than dataset composition, emphasizing the critical role of data in shaping model outcomes.

Our key contributions are:

- Application of Monte Carlo simulations to quantify variability in fairness metrics.
- Aggregation and analysis at the patient level to assess demographic subgroup effects.
- Use of mixed-effects models to link dataset bias with fairness gaps while accounting for hierarchical data structure.

## 2 Related Work

Existing radiography datasets provide general demographic statistics but rarely quantify bias systematically [5]. While prior studies have measured fairness in AI for medical imaging, they typically focus on disparities in performance across demographic groups without statistically relating these disparities to underlying dataset bias [2, 3].

Recent studies, such as [6], discuss metrics to quantify demographic bias in facial expression recognition datasets. These metrics define bias as statistical imbalances in group representation. However, such works do not account for variability or uncertainty in the bias estimates, nor do they extend to more complex datasets such as chest X-rays, where multiple pathologies and demographic correlations exist.

Our work adapts these bias quantification approaches to the medical imaging context, incorporating both bias and its uncertainty. By combining Monte Carlo resampling and mixed-effects modeling, we provide a framework to systematically link demographic bias with model fairness, capturing both the magnitude and reliability of fairness metrics across subgroups.

## 3 Methods

### 3.1 Dataset and Preprocessing

All demographic attributes in this study were derived from the publicly available CheXpert dataset [7], which contains de-identified chest radiographs and metadata collected at Stanford Hospital between 2002 and 2017. CheXpert is released for research under a data use agreement, and all data in this study were used in compliance with its terms.

### 3.1.1 Demographic Data

The CheXpert dataset records gender as Male or Female. In our analysis, we retained these two categories; a single entry with unknown gender was excluded.

Patient age is provided as a continuous variable. For analysis, we stratified age into categorical intervals [0–18, 18–40, 40–60, 60–80, 80+], following the grouping reported in [8].

Race labels are self-reported. Our categorization synthesizes prior approaches in medical imaging fairness research. Following [8], we adopt the subgroups White, Black, and Asian. In line with the work of Seyyed et al. [2], we treat Hispanic as a distinct subgroup rather than aggregating it into Other, motivated by both its substantial representation (n=1,455) and its clinical relevance for health equity research. The final race categories used in this study are Asian, Black, White, Hispanic, Other, and Unknown (for missing or unrecorded entries).

### 3.1.2 Uncertainty Handling

The CheXpert dataset provides target labels with four possible values: positive (1), negative (0), uncertain ($-1$), and unmentioned (blank). For binary classification, we treated unmentioned labels as negative and adopted the U-zeros approach, mapping all uncertain labels to 0 as in [2]. This provides a conservative strategy, as in clinical applications the cost of a false negative (missed diagnosis) is typically considered higher than that of a false positive. Other uncertainty handling strategies, such as U-ones (mapping uncertain labels to 1) or **U-ignore** (excluding uncertain labels), could be investigated in future work to assess the robustness of fairness analyses across different label assumptions.

### 3.1.3 Target Labels

Facial Expression Recognition (FER) datasets typically follow a single-label classification paradigm, where each image is assigned exactly one emotion. In contrast, chest X-ray datasets, such as CheXpert, involve multi-label classification, as a single scan can exhibit multiple co-occurring conditions, such as pneumonia, edema, and pleural effusion. CheXpert contains 14 target labels, reflecting a wide range of thoracic conditions. Existing approaches for multi-label classification fall into two main categories [9]. Binary label approaches treat each disease as a separate, disjoint classification task, ignoring potential correlations between labels. Label correlation approaches leverage dependencies or co-occurrence among diseases to improve predictive performance, reflecting clinical practice where co-occurring conditions inform diagnosis.

For this study, we focus on five target labels, Atelectasis, Cardiomegaly, Consolidation, Edema, and Pleural Effusion, chosen based on their clinical importance, prevalence, and use as competition tasks in CheXpert [7], and adopt a disjoint binary classification setup. This approach simplifies modeling and facilitates fairness evaluation, particularly when examining the variability of model predictions across repeated patient observations and under standard statistical assumptions of independence.

## 3.2 Quantifying Bias

Bias and fairness are closely related but distinct concepts. We differentiate between dataset bias metrics, which capture statistical imbalances in the representation of demographic groups, and model fairness metrics, which quantify performance gaps across those groups. We hypothesize that the variability of dataset bias metrics is related to the variability of model fairness gaps.

### 3.2.1 Dataset Bias Metrics

We selected three demographic bias metrics recommended by Dominguez et al. [6] for use in facial expression recognition datasets, and adapted them to the CheXpert setting: Shannon Evenness Index (SEI), Simpson's Diversity Index (1–D), and Cramer's V ($\phi_C$).

Shannon Evenness Index (SEI) [10] measures the homogeneity of group representation. Values range from 0 (highly imbalanced) to 1 (perfectly balanced), providing a normalized measure of diversity.

Simpson's Diversity Index (1–D) [11] captures the probability that two randomly selected individuals belong to different demographic groups. Like SEI, it characterizes representation bias by quantifying the degree of demographic diversity independent of target labels.

Together, SEI and Simpson's Index summarize representation bias, defined as unequal demographic group distributions in a dataset. These metrics can be applied to any dataset since they do not require information about clinical or target labels.

Cramer's V ($\phi_C$) [12] quantifies associations between a demographic variable and a target label, capturing stereotypical bias. It is derived from the chi-squared statistic, which assumes independence of observations and requires sufficiently large expected frequencies ($\geq 5$ per cell) for validity.

Applying Cramer's V in medical imaging introduces challenges not present in FER data. In CheXpert, the same patient may contribute multiple radiographs (different views or time points), violating independence assumptions and potentially inflating chi-squared statistics. Moreover, demographic features remain constant across repeated observations, further compounding dependence. To mitigate these issues, we restrict our analysis to major demographic categories (gender, age, race) and ensure adequate subgroup sample sizes to preserve statistical validity.

### 3.2.2 Model Fairness Gaps

To assess how dataset bias metrics relate to model behavior, we evaluated fairness gaps using three widely studied criteria. These metrics are standard in the algorithmic fairness literature and capture complementary notions of fairness [13]:

- **Equalized Odds**: Requires both false positive and false negative rates to be similar across demographic groups, ensuring that errors are not disproportionately concentrated in one group.

- **Equal Opportunity**: A relaxed version of Equalized Odds, requiring only that true positive rates are equal across groups, thus focusing specifically on underdiagnosis risks.

- **Demographic Parity**: Requires predicted label distributions to be independent of demographic attributes, regardless of ground-truth labels.

We applied these definitions to a subset of top-performing models from the CheXpert competition leaderboard [14]. Fairness gaps were computed by measuring the absolute differences in metric values across demographic subgroups (gender, age, and race) for each of the five clinically important classification tasks described earlier. To capture variability and reduce sensitivity to sample splits, we applied Monte Carlo resampling to the test dataset of CheXpert, producing a distribution of fairness gap estimates for each model–subgroup combination.

## 3.3 Experimental Design

### 3.3.1 Patient-level Aggregation

To account for multiple images per patient in the dataset, we aggregate image-level labels to a single patient-level label using a maximum rule: a patient is assigned a positive label (1) for a given pathology if at least one of their associated images is positive. This aggregation, applied to the training set, ensures a one-to-one mapping between patients and data points, which is critical for satisfying the assumption of record independence in subsequent statistical analyses.

### 3.3.2 Monte Carlo Simulation

To assess the robustness and statistical variability of both dataset bias and model fairness metrics, we performed patient-level Monte Carlo simulations with 5,000 bootstrap iterations, a standard practice consistent with recommendations in the statistical literature [15].

For dataset bias, the simulation was conducted on the training set using the aggregated patient-level labels. In each iteration, patients were sampled with replacement, and dataset bias metrics such as Simpson's Index and Cramer's V were computed across demographic groups.

For model fairness, the simulation was conducted on the test set, where all images corresponding to each bootstrapped patient were retained. This allowed computation of fairness gaps (Equalized Odds, Equal Opportunity, and Demographic Parity) across models, pathologies, and demographic subgroups.

This two-stage approach generates empirical distributions for each metric, from which means, standard deviations, and 95% confidence intervals can be derived. By explicitly accounting for both sampling variability in the data and variability across model predictions, this procedure provides a rigorous assessment of metric reliability under potentially unstable or unrepresentative data.

### 3.3.3 Mixed-Effects Model

To investigate the relationship between dataset bias metrics and model fairness gaps, we employed a mixed-effects modeling framework [16]. The response variable is the fairness gap for a given metric (Equalized Odds, Equal Opportunity, or Demographic Parity), and the independent variable is the corresponding dataset bias metric (Cramer's V).

The hierarchical nature of the data motivates the use of mixed-effects models. Each leaderboard model provides multiple fairness measurements across pathologies and demographic groups, and each patient contributes multiple images. Mixed-effects models allow us to account for this nested structure by including both fixed effects (capturing systematic differences across demographic groups and pathologies) and random effects (capturing correlations among repeated measurements within the same model and variability across models). The model formula is:

$$\text{Fairness gap} \sim \text{Cramer's V} \times \text{Demographic group} + \text{Pathology} + (1 \mid \text{Model})$$

where $(1 \mid \text{Model})$ denotes a random intercept per model. Interaction terms between Cramer's V and demographic group allow the model to capture group-specific sensitivities to dataset bias.

Prior to modeling, measurements were aggregated at the patient level to prevent skewing by patients with multiple correlated images. We also restricted the analysis to major demographic groups (gender, age, and race) to avoid multicollinearity from small or intersectional subgroups.

This mixed-effects approach provides robust estimates of how variability in dataset bias translates to variability in fairness gaps while properly accounting for the hierarchical structure and repeated measurements. Model convergence and fit were checked for all metrics to ensure reliable inference.

**Limitations** Aggregating images to a patient-level dataframe, particularly when using the maximum value of labels, can lead to loss of detailed information. This simplification may obscure finer-grained associations between labels and demographic attributes and prevents analysis of temporal dynamics across multiple observations for the same patient. As a result, some nuances of bias or fairness variability at the image level could be underestimated.

## 4 Results

### 4.1 Dataset Bias Analysis

We first examined the distribution of image counts per patient in the training dataset. Most patients (72.9%) had between one and three images, whereas a small subset contributed substantially more, resulting in a long-tailed distribution (See Figure 1). This motivates the use of patient-level aggregation to prevent over-representation of patients with many images in subsequent analyses.

### 4.1.1 Representational Bias

We assessed dataset representativeness and subgroup balance using two complementary metrics, Shannon evenness and Simpson diversity indices. Results indicate an even gender distribution (mean evenness = 0.991, 95% CI [0.990, 0.992]; Simpson diversity = 0.494, 95% CI [0.493, 0.495]), consistent with near-equal representation across the two gender categories. Although the Simpson diversity value appears low, this is expected given only two categories, and it actually reflects a near-maximal diversity for gender. For age groups, balance was lower but still relatively high (evenness = 0.824, 95% CI [0.823, 0.826]; diversity = 0.719, 95% CI [0.717, 0.720]). Considering that the maximum possible Simpson diversity for five age categories is 0.8, this indicates that the dataset includes a broad mix of age groups, with only slight overrepresentation of certain brackets. In contrast, race distribution showed the largest imbalance, with lower evenness (0.690, 95% CI [0.686,

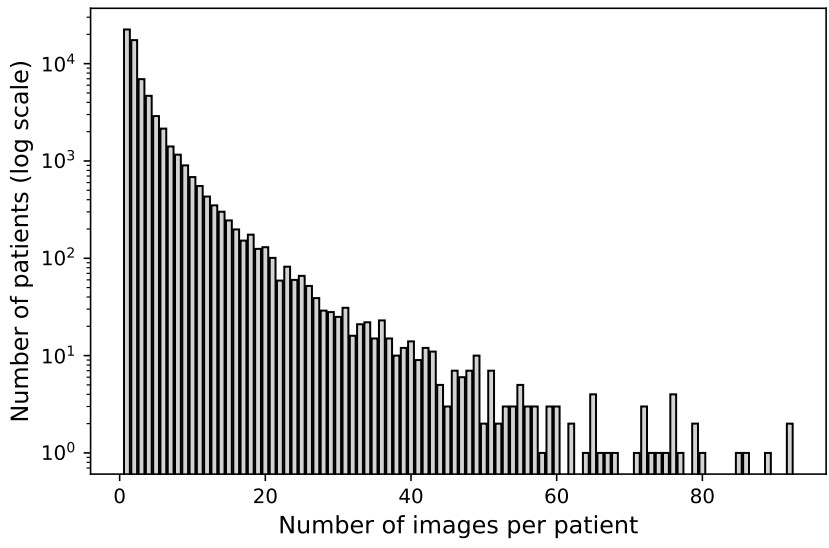

Figure 1: Distribution of image counts per patient in the training dataset.

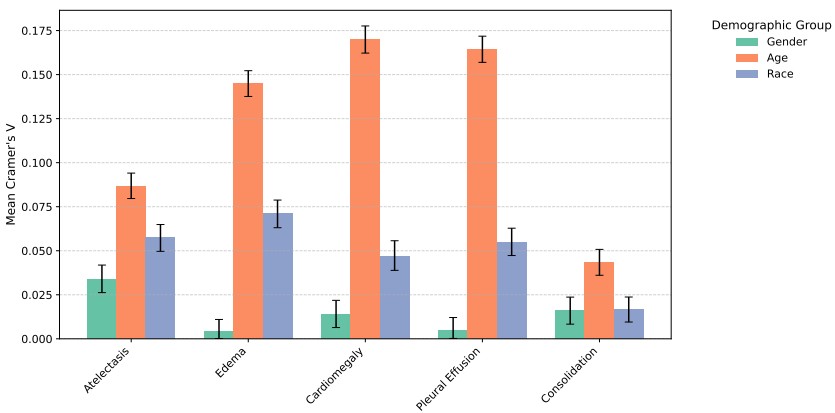

Figure 2: Cramer's V by pathology and demographic group.

0.693]) and diversity (0.640, 95% CI [0.637, 0.643]), reflecting underrepresentation of several racial subgroups.

Standard deviations were consistently small (<0.002) and confidence intervals tight, indicating that the bootstrap estimates are stable and precise. Collectively, these results suggest that while the dataset is well-balanced by gender, age coverage is reasonably broad, and race remains a source of representational bias, which could have implications for fairness evaluations.

### 4.1.2 Stereotypical Bias

Analysis of Cramer's V across pathologies and demographic groups revealed that associations between age and certain conditions were modest but consistent (see Figure 2). For instance, age grouping showed higher Cramer's V for Edema (0.145, 95% CI [0.138, 0.152]) and Cardiomegaly (0.170, 95% CI [0.162, 0.178]), indicating a systematic relationship between patient age and these pathologies in the dataset. Biases across gender and race were generally smaller, with mean Cramer's V values below 0.08. Even modest Cramer's V values, such as 0.14, are meaningful in a dataset of this scale, reflecting subtle but systematic demographic imbalances. Standard deviations were consistently low (<0.004) and confidence intervals very narrow, confirming that these associations are stable and not due to random variation.

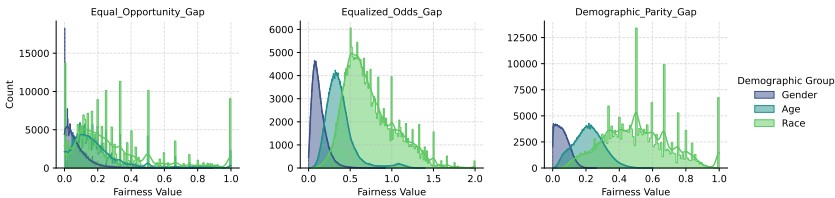

Figure 3: Distribution of fairness metrics by demographic group.

## 4.2 Fairness Gaps Variability

We assessed model fairness across demographic groups using three metrics: Demographic Parity Gap, Equal Opportunity Gap, and Equalized Odds Gap. Results indicate that fairness gaps vary substantially by subgroup. Race exhibits the largest gaps (e.g., mean Equalized Odds Gap = 0.733, std = 0.303), while gender shows the smallest disparities (mean = 0.127, std = 0.086). Age subgroups fall in between (mean = 0.375, std = 0.176). Standard deviations indicate notable variability in fairness outcomes across subgroups, particularly for race. The distributions of these metrics are visualized in Figure 3, which shows histograms for each metric by demographic group. The figure illustrates both the central tendencies and the spread of fairness gaps, confirming that while some disparities are small (e.g., gender), others remain substantial and variable, particularly for race.

## 4.3 Mixed-Effects Model Results

We fitted mixed-effects models to examine the relationship between dataset bias, as measured by Cramer's V, and fairness gaps across three metrics: Equal Opportunity Gap, Equalized Odds Gap, and Demographic Parity Gap. Models included random intercepts for leaderboard models to account for repeated measurements, fixed effects for demographic groups and pathologies, and interactions between Cramer's V and demographics.

For the Equal Opportunity Gap, Cramer's V was negatively associated with fairness for most groups (Coef = -0.659, SE = 0.008, $p < 0.001$), with the interaction term indicating that gender had a stronger sensitivity (Coef = 1.695, SE = 0.038, $p < 0.001$). Pathology effects were modest but significant, with Edema (Coef = 0.062) and Consolidation (Coef = 0.066) showing positive associations. Random effect variance per model was small (0.004), indicating that model-to-model variability contributed little relative to dataset bias.

For the Equalized Odds Gap, associations were weaker but still significant (Cramer's V Coef = -0.057, SE = 0.009, $p < 0.001$), and gender again showed heightened sensitivity (interaction Coef = 0.472, SE = 0.041, $p < 0.001$). Pathology effects varied, with Pleural Effusion showing a notable negative effect (Coef = -0.071).

For the Demographic Parity Gap, Cramer's V exhibited a strong positive effect (Coef = 0.997, SE = 0.004, $p < 0.001$), while the interaction with gender was strongly negative (Coef = -2.903, SE = 0.019, $p < 0.001$). Pathology-specific effects were significant but small (e.g., Pleural Effusion Coef = -0.080). Random effect variance was essentially zero, indicating that the hierarchical model structure did not substantially affect the estimates.

These results indicate that systematic dataset biases, quantified via Cramer's V, are significantly associated with fairness gaps, with certain demographic groups, particularly gender, showing stronger sensitivity. Variability across leaderboard models was minimal, confirming that observed effects are robust to model differences.

## 5 Discussion

Our findings reveal systematic differences in demographic balance across subgroups. Gender is nearly balanced, with a Simpson diversity value close to the theoretical maximum for two groups. In contrast, age distribution skews toward older patients, with younger adults and children underrepresented. Racial diversity is moderate, but the dataset is dominated by White patients, with minority groups (e.g., Native, Hispanic, Black) present in very small numbers. While gender representation is strong,

age and racial composition may introduce fairness concerns, particularly for underrepresented groups. The narrow confidence intervals indicate these imbalances are stable and not due to sampling noise.

Associations between age and certain pathologies, such as Edema and Cardiomegaly, likely reflect both genuine prevalence patterns and dataset-specific collection biases. While such relationships are expected clinically, their persistence in a machine learning dataset raises concerns for reliability; models may inadvertently exploit demographic correlates rather than underlying disease features. Gender and race showed weaker associations, yet even subtle imbalances can propagate through model predictions, especially in high-capacity models. These results underscore the importance of auditing demographic–pathology relationships when building reliable models from clinical data.

Results of fairness gaps variability highlight that even when models are trained on the same dataset, fairness outcomes can vary substantially across demographic groups. The large disparities observed for race suggest persistent underrepresentation or structural imbalances in the data that propagate through model predictions. In contrast, the smaller gaps for gender indicate relatively balanced representation in this subgroup. The high standard deviations for some metrics, particularly for race, indicate that fairness is sensitive to the sampling of patients, emphasizing the importance of robust evaluation methods.

The mixed-effects models demonstrate that even modest dataset biases can propagate into measurable fairness gaps. Gender consistently shows heightened sensitivity to Cramer's V across all metrics, suggesting that models trained on the dataset may systematically disadvantage one gender if biases are present. Pathology-specific effects, while smaller, indicate that dataset composition interacts with disease prevalence to influence fairness outcomes. Random effect variance was minimal across leaderboard models, implying that differences in model architecture or training approach contribute less to fairness variability than underlying dataset biases. This emphasizes the importance of auditing dataset representativeness when developing and deploying models.

Taken together, these findings highlight that subtle but systematic demographic imbalances, even when small in magnitude, can significantly affect fairness and reliability. Careful dataset auditing and bias mitigation are therefore essential for medical ML. In particular, fairness metrics proved unstable in small subgroups, where a single misclassification could dramatically shift results. This volatility poses a fundamental challenge; fairness evaluation is not only about bias but also about the reliability of subgroup estimates.

Finally, our study treated each scan independently, ignoring longitudinal dependencies from repeated measurements. Incorporating temporal structure would enable more robust fairness assessments over time. Future work should also move beyond static evaluations. We envision *a Fairness Volatility Index* to systematically quantify subgroup stability across datasets and over time, aligning fairness evaluation with the broader goals of reliable machine learning under real-world data limitations.

# 6   Conclusion

Our study shows that fairness metrics in medical imaging are not only influenced by dataset bias but also by the reliability of subgroup estimates. Even when dataset bias, as measured by Cramer's V, is stable, fairness gaps can fluctuate substantially, especially in small or underrepresented subgroups where a single misclassification can drastically change the metric. Mixed-effects modeling revealed that fairness gaps are significantly associated with dataset bias, with demographic groups such as gender showing greater sensitivity. However, random effects across models were small, indicating that instability arises more from data distribution than from model variation.

These findings underscore the need for fairness evaluations that account for subgroup size and reliability, paving the way for methods that measure not only bias but also its stability.

**Acknowledgments and Disclosure of Funding**

The authors acknowledge Gemini for assisting with drafting and revising text and refining methodological details, including exploring statistical approaches such as patient-level bootstrapping and mixed-effects modeling. All scientific content and conclusions are the responsibility of the authors.

This work is funded by Project PID2023-149079OB-I00 funded by MI-CIU/AEI/10.13039/501100011033 and by ERDF/EU.

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
