# OpenReview forum: "Beyond Static Bias: Quantifying Fairness Variability in CheXpert"
_NeurIPS.cc/2025/Workshop/Reliable_ML — NeurIPS 2025 - Reliable ML Workshop_

### Official Review · Reviewer_g4F9 · 2025-09-20
**Important discussion, but not very clear central position**

**Rating:** 6
**Confidence:** 2

**Review:**

# Summary

This paper addresses the limitations of using static, point-estimate metrics for evaluating fairness in medical AI models. The authors focus on the CheXpert dataset (a chest radiographs dataset) and use three metrics to measure dataset bias and three criteria to measure model fairness gap. They use bootstrapping (patient-level Monte Carlo simulations) to generate distributions for both dataset bias metrics and model fairness gaps. They then statistically link these quantities by using a mixed-effects model. The key finding is that even when the dataset bias is modest and stable, the model fairness can be highly volatile and unpredictable.

# Strengths

* The paper is very relevant to the reliable machine learning workshop theme. It discusses an important topic for ensuring fairness and reliability in ML methods.

* The central finding is important and the methodology is sound.

# Weaknesses

* There are many missing details regarding what models (and how many) were chosen from the CheXpert leaderboard, which is an important information. For example, could it be that they all have very similar architectures so then this explains why "model variability contributes less to fairness variability than dataset composition" ?

* I think that in the current presentation of your paper, your contribution is not clear enough. You need to motivate the use of bootstrapping over static point-estimate metrics more if you want your paper to actually be impactful.

# Suggestions for Authors

* Please give very detailed descriptions of your experiments and methodological decisions. These are essential for validating your claims. What models did you use? Why did you use Cramer's V as the sole predictor for dataset bias in the mixed-effects model? What would be the results if you tried a few different patient-level aggregation methods other than just the `max` rule?

* This might sound cliche but: you should avoid the passive voice when writing. It would be a lot easier to read your paper if it made less use of passive voice